# Household structure and its association with sexual risk behaviours and sexual health outcomes: evidence from a British probability sample survey

Tyrone J Curtis,[1] Nigel Field,[2] Soazig Clifton,[1] Catherine H Mercer[1]

¹Centre for Population Research in Sexual Health and HIV, Institute for Global Health, University College London, London, UK
²Centre for Molecular Epidemiology and Translational Research, Institute for Global Health, University College London, London, UK

**Correspondence to**
Tyrone J Curtis;
tyrone.curtis@ucl.ac.uk

## ABSTRACT

**Objectives** Economic and social changes over the last 20 years have led to changes in the living situations of young people in Britain. A person's home-life context might influence their sexual behaviour, with implications for their sexual healthcare needs; we investigated this hypothesis.

**Methods** Britain's third National Survey of Sexual Attitudes and Lifestyles, a probability sample survey undertaken in 2010–2012, interviewed 15 162 men and women aged 16–74 years in Britain (with 3869 aged 16–24 years). We examined household structure by gender and age group. We then focused on sexually experienced young people (aged 16–24 years), and used multivariable models to explore associations between household structure, sexual risk behaviours and sexual health outcomes, independent of confounders including age, relationship status, employment and area of residence.

**Results** Young people were most likely to be living with parents (women 57.1% (95% CI 54.5% to 59.6%) and men 68.7% (95% CI 65.4% to 71.8%)) or non-relatives (women 10.5% (95% CI 8.5% to 12.9%) and men 12.6% (95% CI 10.1% to 15.6%)). Among the 81.3% of young people who were sexually experienced, compared with young women living with parents (reference category), young women living alone or with non-relatives had a higher likelihood of reporting ≥2 sexual partners (adjusted OR 1.54 (95% CI 1.03 to 2.31); 1.76 (95% CI 1.03 to 3.00), respectively). Women living alone were also more likely to have had unsafe sex (2.04 (95% CI 1.38 to 3.02)). Despite these differences in sexually transmitted infection (STI) risk, there was no difference in sexual healthcare-seeking behaviour. Young men and women living with partners reported lower levels of sexual risk behaviours.

**Conclusions** Our study suggests household structure may influence the sexual behaviour of young people in Britain. Given changes in their living arrangements, the role of household structure in sexual health research should be further investigated, and also considered as a possible marker for STI risk in clinical consultations.

## INTRODUCTION

Young people in the UK are increasingly spending their early adulthood living with parents.[1] One reason for this may be the increased uptake of higher education, with twice as many students in 2014 living at

### Strengths and limitations of this study

► This is the first study to investigate the association between household structure and sexual behaviour and sexual health among young people, a population group that bears a disproportionate burden of sexual ill-health.
► The study analysed data from a large, national probability sample survey of sexual health and so results can be considered as broadly representative of the British general population.
► The comprehensive nature of the survey enabled us to examine a broad range of sexual risk behaviours and sexual health outcomes.
► The numbers of participants from ethnic minority groups were relatively small, and so it was not possible to adjust for ethnicity in analyses.
► The survey's sampling frame—private households—means that those living in institutions (including some students) or who are homeless are excluded.

home for the duration of their studies than in 1996.[2] Those students who do move away from home are increasingly likely to return to the parental home on completion of their studies, if only for a brief period.[3] Traditionally, a key driver for leaving the parental home has been marriage or, more recently, to cohabit with a partner. However, young people are deferring the age at which they first move in with a partner, and so tend not to form new family units until later in their 20s and early 30s.[2] The living arrangements of young people are also heavily dependent on economic circumstances. The economic uncertainty caused by the financial crisis of 2007–2008 and the recession that followed disproportionately affected young people, resulting in high levels of youth unemployment,[4] a factor known to increase the proportion of young people living at home with parents.[5] Restrictions in access to the welfare system and housing market have

also made it more likely for young people outside of family households to be living in rented accommodation, often sharing with friends or other non-related adults.[6] Overall, more young people are living at home with their parents, and those who do leave home are more likely to be living for a time outside of a family household.

It is plausible that these changes might affect sexual behaviour, and by implication, sexual health, for example, due to changes in factors identified, through a review guided by the integrative model of behavioural prediction,[7] as acting as constraints on young people's sexual behaviour such as parental monitoring and supervision.[8] Associations have been found between higher-risk sexual behaviour and family structure, that is, the presence of one or both natural parents in the household,[9] as well as levels of parental monitoring,[10] however these studies have usually focused on young people below the age of 18 years. Across the life course, relationship and/or cohabitation status have often been shown to be associated with differences in sexual behaviour[11 12] and sexual health outcomes.[13] However, to date, there has been no investigation into the variations in sexual behaviour and sexual health outcomes according to different types of household structure more broadly, that is, the composition of individuals' households based on their relationships to all other household members, and not just the presence of partners or parents.

Sexual health surveys such as Britain's National Survey of Sexual Attitudes and Lifestyles (Natsal) routinely collect information on participants' household structure. In this study, we used Natsal-3's nationally representative data to explore the distribution of household structures for the British population aged 16–74 years, and then investigated the independent associations between household structure and the sexual health of young people, specifically young adults aged 16–24 years as the group that experiences the greatest social and sexual transitions[14] and among the highest burden of poor sexual health, including sexually transmitted infections (STIs).[15 16]

## METHODS
### Participants and procedures
Full details of the methods of Natsal-3 have been reported elsewhere.[17 18] Briefly, Natsal-3 was a survey of 15 162 men and women aged 16–74 years resident in Britain, undertaken from 2010 to 2012, and recruited using a multistage, clustered, stratified probability sample design. The response rate was 57.7%, with a cooperation rate (the proportion of interviews completed from eligible addresses for which contact was made) of 65.8%.[18] Included in the sample were 3869 men and women aged 16–24 years (1729 men). Participants provided oral informed consent for interviews.

Participants were interviewed in their own homes by professional interviewers using a combination of a face-to-face computer-administered personal interview and a computer-assisted self-interview (CASI). Most questions relating to general health and well-being were collected face-to-face. More sensitive questions about participants' sexual behaviour and sexual health, including questions about STI testing and diagnoses, and sexual health clinic attendance, were asked via CASI. Participants were routed into the CASI if they reported sex with an opposite-sex partner and/or reported any sexual experience with someone of the same sex (both since age 13 years), or if they refused to answer this question. Participants reporting no previous sexual experience did not complete the CASI.

After completing the CASI, participants were asked to provide information on up to 11 additional household members, including their age, sex and relationship to the participant (partner, son or daughter (including step, foster or in-law), parent (including step, foster or in-law), brother or sister (including step, half, foster or in-law), grandchild, grandparent, other relative, or non-relative). Information on other members of participants' households was used to devise six household structure categories (table 1). In the criteria below, 'children' and 'grandchildren' refer to those of the participant, regardless of age and including step, foster or in-law.

### Patient and public involvement
As an unfunded secondary analysis of Natsal-3 data, this particular paper did not entail patient and public involvement. However, public engagement has been a key component of each phase of the (to date) three rounds of the Natsal study, including public consultation exercises to inform new topic areas, cognitive interviews to inform question wording, and interactive public engagement events (including via digital media) to disseminate findings.

### Statistical analysis
All analyses were carried out using the complex survey functions of Stata (V.15.1) to incorporate the weighting, clustering and stratification of Natsal-3.[18] We used an α of 0.05 in all analyses.

We first calculated the distribution of household structure by gender and age group for all participants aged 16–74 years (0.6% non-response). To examine how the reporting of sexual risk behaviours and sexual health outcomes by young people varied by household structure category, we used data for 1717 women and 1361 men aged 16–24 years who reported having had at least one sexual partner in their lifetime (hereon 'sexually experienced young people'). We calculated descriptive statistics for each reported outcome, and used multivariable logistic regression to calculate adjusted ORs (AORs) to compare participants living with parents or other relatives with those living in other household structure categories.

**Table 1** Household structure categories derived from Natsal-3 questionnaire responses about other members of participants' households

| Household structure category | Criteria |
| --- | --- |
| Lives with parents or other relatives* | Lives with parents or other relatives (including siblings), but not with a partner or children/grandchildren of their own. Household may include non-relatives. |
| Lives alone | No other household members. |
| Lives with non-relatives only | Does not live with parents, other relatives, a partner or children/grandchildren of their own. |
| Lives with a partner, not with children/grandchildren | Lives with a partner but not with children/grandchildren of their own. Household may include other members. |
| Lives with a partner and children/grandchildren | Lives with a partner and children/grandchildren of their own. Household may include other members. |
| Lives with children/grandchildren, not with a partner | Lives with children/grandchildren of their own but does not live with a partner. Household may include other members. |

*Participants living with other relatives were grouped with those living with parents recognising that some young people may not live with their parents, for a variety of reasons, and may instead be living with other caretakers such as grandparents or even older siblings, and their relationship to these caretakers may have a similar influence on behaviour. Furthermore, only 5.4% (n=164) of all participants (and 3.2% (n=79) of participants aged 16–24 years) in this category did not live with parents.

We examined a number of sexual behaviours linked to STI acquisition[19] and sexual health outcomes, in recognition of the multidimensional nature of sexual health.[20] Sexual intercourse was defined as reporting vaginal, oral or anal sex with a partner of the opposite sex, or oral or anal sex or genital contact with a partner of the same sex. Unsafe sex was defined as reporting two or more partners of the opposite or same sex with no condom use in the past year, or no condom use at first sex with one or more new partners of the opposite or same sex in the past year. STI diagnosis in the past year was defined as reporting diagnosis with one of the following STIs in the past year: trichomoniasis, gonorrhoea, chlamydia, syphilis, non-specific or non-gonococcal urethritis, genital warts, or herpes. Other outcomes examined include measures of recency of sex and sexual frequency, reporting two or more partners in the past year, reporting having tested for chlamydia in the past year, and reporting having attended a genitourinary medicine (GUM) clinic in the past year.

In multivariable models, we adjusted for factors in Natsal-3 hypothesised to be associated with both household structure and sexual health. We first adjusted for age and relationship status (categorised as in a steady relationship or not), as the factors most strongly correlated with sexual behaviour, and then further adjusted for area of residence (rural, Greater London or other urban) and socioeconomic status. To account for variation by socioeconomic status while also recognising the difficulty in capturing social class in young people,[21] we used the National Statistics Socio-Economic Classification[22] (categorised here as in employment, in full-time education or neither). Relationship status was derived from responses to questions about legal marital status as well as the participant's relationship to their most recent sexual partner(s), categorising participants as either in a steady relationship (married, in a civil partnership, in a steady relationship but not married or in a civil partnership) or not. Due to low

numbers of participants of non-white ethnicity, it was not feasible to take account of ethnicity in our models.

## RESULTS

### Variations in household structure by gender and age group

The majority of people aged 16–74 years in Britain in 2010–2012 lived with a partner or a partner and children, with the proportions similar for men (62.6% (61.2%–64.0%)) and women (60.7% (59.6%–61.9%)) (figure 1). However, there were differences by gender for those not living with partners. More women than men were living with children but not with a partner (11.9% (11.1%–12.7%) vs 1.8% (1.4%–2.2%); p<0.001), while more men than women lived with their parents (11.2% (10.6%–11.8%) vs 16.0% (15.2%–17.0%); p<0.001). A greater proportion of older women lived alone; 30.4% (27.6%–33.4%) of women aged 65 years or older lived alone, compared with 19.3% (16.8%–22.1%) of men of the same age (p<0.001).

Among people aged 16–24 years, fewer women reported living in their parents' home than men; 57.1% (54.5%–59.6%) of women lived with parents or other relatives, compared with 68.7% (65.4%–71.8%) of men of similar age (p<0.001). More women were living in their own family households; 25.6% (23.6%–27.7%) lived with a partner and/or children, compared with 11.1% (9.5%–13.0%) of men (p<0.001). Similar proportions of young men and women reported living in non-family households; 7.6% (6.2%–9.4%) of men and 6.9% (5.9%–8.1%) of women lived alone, and 12.6% (10.1%–15.6%) of men and 10.5% (8.5%–12.9%) of women lived only with non-relatives.

### Sample characteristics of sexually experienced young people (aged 16–24 years) by household structure

Focusing on sexually experienced young people (80.9% of young women, 81.7% of young men), 50.8% of sexually experienced young women lived with parents or other

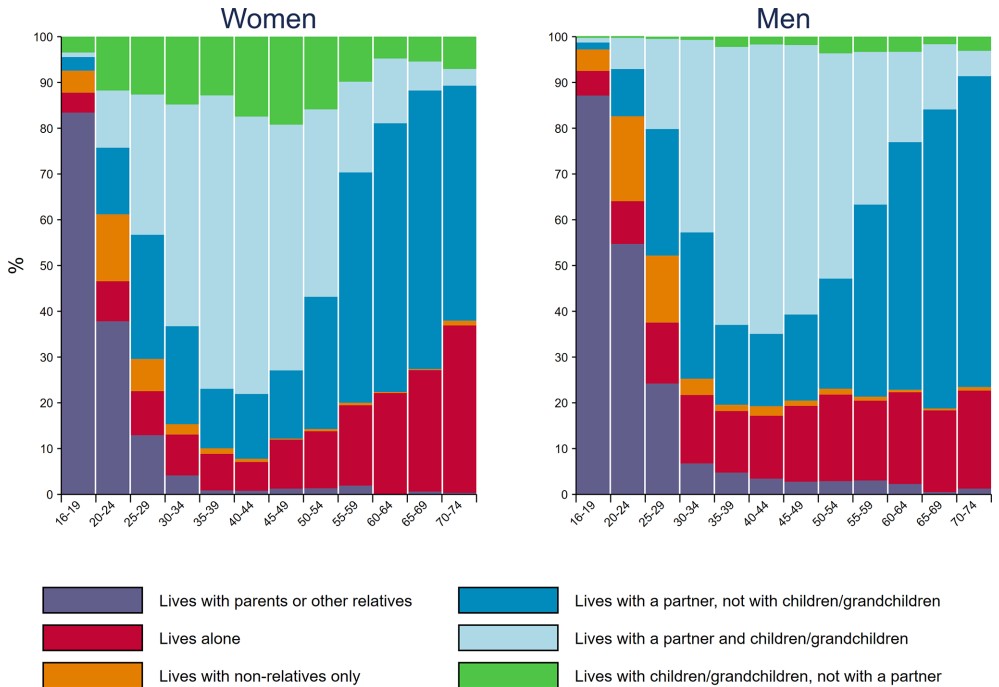

**Figure 1** Household structure by gender and age group among the resident British population aged 16–74 years.

relatives, 7.4% lived alone, 10.6% lived with non-relatives only, 11.9% lived with a partner but not with children, 9.4% lived with a partner and children, and 9.9% lived with children but not with a partner. Among sexually experienced young men, 65.2% lived with parents or other relatives, 8.3% lived alone, 13.1% lived with non-relatives only, 7.9% lived with a partner but not with children, and 5.5% lived with a partner and children.

The majority of sexually experienced young people in our sample was of white ethnicity (88%), and lived in Greater London or other urban areas (table 2). Around 55% were in employment, with a further 37.5% in full-time education. A greater proportion of women than men were in steady relationships (65.6% vs 52.5%, p<0.001). The majority of young people living only with non-relatives were in full-time education (69.5% of women, 76.7% of men), while the majority of those living with partners or partners and children were in employment.

### Variations in sexual risk behaviours and sexual health outcomes reported by sexually experienced young people (aged 16–24 years) by household structure

Compared with young women living with parents or other relatives, more of those living alone, living with partners but not with children, or living with partners and children reported having had sex in the past 4 weeks (weighted prevalence 74.0% vs 83.7%, 97.7% and 90.2%, respectively) (table 3). After adjusting for age, relationship status, area of residence and employment/student status, women living alone and women living with partners but not with children remained more likely to report having had recent sex than their counterparts living with parents or other relatives (AOR 1.84 (95% CI 1.11 to 3.04) and 3.32 (1.13–9.74), respectively). Similarly, women living

alone (45.5% vs 40.7%; AOR 1.54 (1.03–2.31)) or with non-relatives (47.5% vs 40.7%; AOR 1.76 (1.03–3.00)) were more likely to report two or more partners in the past year. Women living alone were also more likely to have had unsafe sex in the past year (44.2% vs 32.7%; AOR 2.04 (1.38–3.02)).

Although women living alone and women living with non-relatives were more likely to report sexual risk behaviours, these groups were not more likely to report chlamydia testing or GUM clinic attendance in the past year than women living with parents. However, women living with children but not with a partner were more likely to report having tested for chlamydia in the past year (66.2% vs 55.8%; AOR 1.54 (1.08–2.21)). Women living with partners were less likely than women living with parents to report sexual health service usage (chlamydia testing or GUM clinic attendance).

After adjusting for age, relationship status, employment/student status and area of residence, few associations were found between household structure and sexual risk behaviours and sexual health outcomes for men (table 4). Compared with men living with parents or other relatives, men living with a partner and no children (11.9% vs 48.9%; AOR 0.21 (0.10–0.46)) and men living with a partner and children (2.8% vs 48.9%; AOR 0.04 (0.01–0.18)) were significantly less likely to report having had two or more partners in the past year. Men living with a partner and children were also less likely to have had unsafe sex in the past year (9.6% vs 31.3%; AOR 0.25 (0.11–0.57)) and to report testing for chlamydia in the past year (20.3% vs 37.1%; AOR 0.47 (0.24–0.91)).

**Table 2** Sociodemographic characteristics and household structure of sexually experienced women and men aged 16–24 years

| Women | Lives with parents or other relatives (%) | Lives alone (%) | Lives with non-relatives only (%) | Lives with a partner, not with children (%) | Lives with a partner and children (%) | Lives with children, not with a partner (%) | Total sample (%) |
|---|---|---|---|---|---|---|---|
| **Age** | | | | | | | |
| 16–17 years | 25.7 | 6.2 | 0.9 | 1.0 | 0.4 | 3.1 | 14.1 |
| 18–19 years | 29.8 | 18.2 | 15.4 | 11.4 | 5.1 | 14.8 | 21.4 |
| 20–24 years | 44.5 | 75.6 | 83.7 | 87.7 | 94.5 | 82.1 | 64.5 |
| **Ethnicity** | | | | | | | |
| White | 89.4 | 82.6 | 82.5 | 89.6 | 85.6 | 85.4 | 87.5 |
| Mixed | 3.0 | 5.0 | 7.2 | 3.0 | 4.3 | 8.4 | 4.2 |
| Asian/Asian British | 2.7 | 5.9 | 2.7 | 2.3 | 6.5 | 1.1 | 3.1 |
| Black/Black British | 4.0 | 5.2 | 4.4 | 1.1 | 2.8 | 5.2 | 3.8 |
| Other | 0.9 | 1.3 | 3.3 | 4.0 | 0.8 | 0.0 | 1.4 |
| **Employment/student status** | | | | | | | |
| In employment | 47.7 | 54.9 | 27.0 | 76.7 | 82.6 | 68.8 | 54.9 |
| In full-time education | 46.2 | 31.5 | 69.5 | 17.4 | 2.4 | 10.4 | 36.5 |
| Not in employment or education | 6.1 | 13.7 | 3.5 | 5.9 | 15.0 | 20.8 | 8.7 |
| **Area of residence** | | | | | | | |
| Rural | 20.7 | 11.3 | 3.7 | 24.3 | 14.0 | 15.9 | 17.5 |
| Greater London | 11.8 | 20.3 | 9.7 | 7.8 | 4.9 | 15.5 | 11.4 |
| Other urban | 67.5 | 68.4 | 86.6 | 68.0 | 81.1 | 68.6 | 71.0 |
| **Relationship status** | | | | | | | |
| Not in a steady relationship | 43.6 | 38.2 | 49.3 | 0.0 | 0.0 | 42.9 | 34.3 |
| In a steady relationship | 56.4 | 61.8 | 50.7 | 100.0 | 100.0 | 57.1 | 65.6 |
| **Denominators** | | | | | | | |
| Unweighted total (N) | 872 | 178 | 101 | 171 | 149 | 246 | 1717 |
| Weighted total (N) | 486 | 71 | 101 | 114 | 90 | 94 | 956 |
| Total (%) (weighted) | 50.8 | 7.4 | 10.6 | 11.9 | 9.4 | 9.9 | |

| Men | Lives with parents or other relatives (%) | Lives alone (%) | Lives with non-relatives only (%) | Lives with a partner, not with children (%) | Lives with a partner and children (%) | Lives with children, not with a partner (%) | Total sample (%) |
|---|---|---|---|---|---|---|---|
| **Age** | | | | | | | |
| 16–17 years | 21.7 | 2.8 | 0.0 | 0.0 | 2.2 | – | 14.5 |
| 18–19 years | 26.3 | 25.0 | 17.0 | 10.2 | 8.7 | – | 22.7 |
| 20–24 years | 52.0 | 72.2 | 83.0 | 89.8 | 89.1 | – | 62.8 |
| **Ethnicity** | | | | | | | |
| White | 86.7 | 89.7 | 83.4 | 98.0 | 96.0 | – | 87.9 |
| Mixed | 3.8 | 0.9 | 3.7 | 0.0 | 0.0 | – | 3.0 |
| Asian/Asian British | 5.2 | 3.4 | 3.7 | 1.0 | 3.1 | – | 4.4 |
| Black/Black British | 3.2 | 2.1 | 7.3 | 1.0 | 0.0 | – | 3.3 |
| Other | 1.1 | 3.8 | 1.9 | 0.0 | 1.0 | – | 1.3 |
| **Employment/student status** | | | | | | | |
| In employment | 54.9 | 58.5 | 23.3 | 85.7 | 94.3 | – | 55.7 |
| In full-time education | 37.5 | 33.4 | 76.7 | 12.5 | 4.2 | – | 38.5 |

Continued

**Table 2**  Continued

| Men | Lives with parents or other relatives (%) | Lives alone (%) | Lives with non-relatives only (%) | Lives with a partner, not with children (%) | Lives with a partner and children (%) | Lives with children, not with a partner (%) | Total sample (%) |
|---|---|---|---|---|---|---|---|
| Not in employment or education | 7.6 | 8.1 | 0.0 | 1.8 | 1.5 | – | 5.8 |
| Area of residence | | | | | | | |
| Rural | 22.2 | 9.3 | 4.3 | 13.5 | 23.0 | – | 18.2 |
| Greater London | 13.6 | 7.8 | 15.6 | 5.2 | 3.0 | – | 12.1 |
| Other urban | 64.1 | 82.9 | 80.1 | 81.3 | 74.1 | – | 69.7 |
| Relationship status | | | | | | | |
| Not in a steady relationship | 57.2 | 43.2 | 51.2 | 0.0 | 0.0 | – | 47.5 |
| In a steady relationship | 42.9 | 56.8 | 48.8 | 100.0 | 100.0 | – | 52.5 |
| Denominators | | | | | | | |
| Unweighted total (N) | 941 | 156 | 106 | 89 | 69 | 4 | 1365 |
| Weighted total (N) | 648 | 82 | 130 | 79 | 55 | – | 994 |
| Total (%) (weighted) | 65.2 | 8.3 | 13.1 | 7.9 | 5.5 | - | |

Figures presented are column percentages. Due to low numbers (n=4), statistics are not presented for young men living with children but not with a partner.

## DISCUSSION

Using national probability survey data, we identified six distinct types of household structures in the British population, which differed by gender and age group. Of note, women were more likely than men to live with children but not with a partner, and at older ages, to live alone. We also found that young men were more likely than young women to live with parents, while young women were more likely than young men to live in new family households with a partner and/or children. We found for both young men and women that household structure was associated with reporting sexual risk behaviours. Young men and women cohabiting with partners, with or without children, were less likely to report these behaviours than their counterparts who lived with their parents or other relatives, while young women living alone or with non-relatives were more likely to report these behaviours. Importantly, these associations were independent of age and relationship status. Although young women living outside of family households were more likely to report sexual behaviours linked to STI acquisition, these women were not more likely to report GUM clinic attendance or having tested for chlamydia in the past year. Further adjustment for the number of partners in the past year (not shown) did not remove this association.

Our finding that young women living alone or with non-relatives were more likely than women living with parents to report sexual risk behaviours was similar to those of a number of studies of young people in USA which showed that, when compared with young people living at home with parents, those living away from home were more likely to report casual sex,[23 24] more casual sex partners,[25] inconsistent condom use, and higher-risk sex (including condomless casual sex),[23] and more likely to

report having sex while drunk or high.[24] Lyons et al speculated that these differences were due to variations in parental monitoring which allow more opportunities for casual sex,[25] and indeed freedom from parental supervision was cited as a reason for increased sexual activity in interviews with UK university students.[26] Our finding of associations between living away from parents and higher reporting of sexual risk behaviours in young women but not in young men is supported by evidence from a large study of Scottish teenagers which found that a low level of parental monitoring was associated with early sexual activity for both men and women, but additionally for women with higher numbers of sexual partners, and inconsistent condom and contraceptive use.[10] Similar associations have been found between parental knowledge of other risk behaviours, including drinking and drug use,[27] and other studies have found associations between living away from parents and increased alcohol use and drug use in young people, especially students.[28 29] Additionally, we found associations with sexual risk behaviours for women living alone that were not found for women living with non-relatives, perhaps indicating that for young women, living with others—regardless of who—moderates sexual risk behaviour.

The transition between childhood, marked by dependence on parents or other family members, to adulthood, signified by independence, partnership and parenthood, is recognised as a period of greater risk-taking.[30] A young person's transition route, including the age at which this transition begins, as well as the duration, is dependent on many contextual factors including their gender, educational attainment and socioeconomic status, as well as external factors such as labour market conditions and public policy,[31] all of which are also associated with

**Table 3** Sexual risk behaviours and sexual health outcomes in sexually experienced women aged 16–24 years

| | Denominator (unweighted) | Denominator (weighted) | Prevalence (%) | OR | 95% CI | Partially adjusted OR* | 95% CI | Fully adjusted OR† | 95% CI |
|---|---|---|---|---|---|---|---|---|---|
| Any sexual intercourse in the past 4 weeks | | | | p<0.001 | | p=0.02 | | p=0.02 | |
| Lives with parents or other relatives | 849 | 472 | 74.0 | 1.00 | | 1.00 | | 1.00 | |
| Lives alone | 176 | 70 | 83.7 | 1.80 | (1.18 to 2.75) | 1.87 | (1.15 to 3.06) | 1.84 | (1.11 to 3.04) |
| Lives with non-relatives only | 96 | 95 | 71.8 | 0.89 | (0.54 to 1.47) | 0.98 | (0.51 to 1.88) | 0.99 | (0.51 to 1.90) |
| Lives with a partner, not with children | 169 | 112 | 97.7 | 14.81 | (5.37 to 40.86) | 3.34 | (1.14 to 9.78) | 3.32 | (1.13 to 9.74) |
| Lives with a partner and children | 147 | 89 | 90.2 | 3.24 | (1.76 to 5.99) | 0.74 | (0.37 to 1.45) | 0.69 | (0.35 to 1.38) |
| Lives with children, not with a partner | 241 | 93 | 74.1 | 1.00 | (0.68 to 1.47) | 1.02 | (0.64 to 1.61) | 0.96 | (0.60 to 1.53) |
| Four or more occasions of sexual intercourse in the past 4 weeks | | | | p<0.001 | | p=0.41 | | p=0.32 | |
| Lives with parents or other relatives | 810 | 453 | 45.3 | 1.00 | | 1.00 | | 1.00 | |
| Lives alone | 158 | 63 | 53.2 | 1.37 | (0.94 to 2.00) | 1.35 | (0.87 to 2.11) | 1.41 | (0.89 to 2.23) |
| Lives with non-relatives only | 90 | 90 | 44.5 | 0.97 | (0.62 to 1.52) | 1.14 | (0.71 to 1.84) | 1.21 | (0.75 to 1.96) |
| Lives with a partner, not with children | 151 | 101 | 67.2 | 2.47 | (1.70 to 3.61) | 1.08 | (0.71 to 1.66) | 1.03 | (0.68 to 1.58) |
| Lives with a partner and children | 137 | 83 | 60.9 | 1.88 | (1.28 to 2.78) | 0.84 | (0.53 to 1.31) | 0.80 | (0.50 to 1.26) |
| Lives with children, not with a partner | 224 | 86 | 40.5 | 0.82 | (0.59 to 1.15) | 0.80 | (0.52 to 1.25) | 0.81 | (0.52 to 1.27) |
| Two or more partners‡ in the past year | | | | p<0.001 | | p<0.001 | | p<0.001 | |
| Lives with parents or other relatives | 866 | 483 | 40.7 | 1.00 | | 1.00 | | 1.00 | |
| Lives alone | 176 | 70 | 45.5 | 1.22 | (0.84 to 1.77) | 1.47 | (1.00 to 2.17) | 1.54 | (1.03 to 2.31) |
| Lives with non-relatives only | 99 | 99 | 47.5 | 1.32 | (0.81 to 2.17) | 1.50 | (0.91 to 2.50) | 1.76 | (1.03 to 3.00) |
| Lives with a partner, not with children | 169 | 112 | 9.9 | 0.16 | (0.09 to 0.27) | 0.29 | (0.16 to 0.52) | 0.28 | (0.16 to 0.49) |
| Lives with a partner and children | 148 | 90 | 6.5 | 0.10 | (0.05 to 0.20) | 0.19 | (0.09 to 0.38) | 0.17 | (0.09 to 0.35) |
| Lives with children, not with a partner | 245 | 94 | 37.3 | 0.87 | (0.63 to 1.20) | 1.03 | (0.72 to 1.48) | 1.00 | (0.69 to 1.43) |
| Had unsafe sex in the past year§ | | | | p<0.001 | | p<0.001 | | p<0.001 | |
| Lives with parents or other relatives | 863 | 482 | 32.7 | 1.00 | | 1.00 | | 1.00 | |
| Lives alone | 176 | 70 | 44.2 | 1.63 | (1.12 to 2.38) | 2.06 | (1.39 to 3.03) | 2.04 | (1.38 to 3.02) |
| Lives with non-relatives only | 98 | 96 | 26.6 | 0.75 | (0.44 to 1.28) | 0.93 | (0.54 to 1.60) | 1.07 | (0.60 to 1.92) |
| Lives with a partner, not with children | 169 | 112 | 12.1 | 0.28 | (0.17 to 0.46) | 0.49 | (0.29 to 0.83) | 0.48 | (0.29 to 0.81) |
| Lives with a partner and children | 149 | 90 | 14.9 | 0.36 | (0.22 to 0.58) | 0.65 | (0.38 to 1.10) | 0.62 | (0.36 to 1.06) |
| Lives with children, not with a partner | 246 | 94 | 35.8 | 1.15 | (0.83 to 1.59) | 1.49 | (1.04 to 2.12) | 1.41 | (0.98 to 2.04) |
| Diagnosed with an STI in past year¶ | | | | p=0.19 | | p=0.19 | | p=0.13 | |
| Lives with parents or other relatives | 869 | 484 | 5.1 | 1.00 | | 1.00 | | 1.00 | |

Continued

**Table 3** Continued

| | Denominator (unweighted) | Denominator (weighted) | Prevalence (%) | OR | 95% CI | Partially adjusted OR* | 95% CI | Fully adjusted OR† | 95% CI |
|---|---|---|---|---|---|---|---|---|---|
| Lives alone | 178 | 71 | 5.9 | 1.17 | (0.53 to 2.57) | 1.03 | (0.43 to 2.49) | 1.05 | (0.41 to 2.68) |
| Lives with non-relatives only | 101 | 101 | 4.8 | 0.93 | (0.21 to 4.04) | 0.80 | (0.17 to 3.84) | 0.94 | (0.21 to 4.17) |
| Lives with a partner, not with children | 170 | 113 | 3.3 | 0.64 | (0.26 to 1.53) | 0.51 | (0.17 to 1.54) | 0.51 | (0.17 to 1.49) |
| Lives with a partner and children | 149 | 90 | 0.8 | 0.16 | (0.04 to 0.69) | 0.12 | (0.02 to 0.66) | 0.12 | (0.02 to 0.61) |
| Lives with children, not with a partner | 244 | 94 | 4.3 | 0.83 | (0.41 to 1.65) | 0.70 | (0.30 to 1.64) | 0.67 | (0.27 to 1.67) |
| Tested for chlamydia in the past year | | | | p<0.001 | | p<0.001 | | p<0.001 | |
| Lives with parents or other relatives | 870 | 485 | 55.8 | 1.00 | | 1.00 | | 1.00 | |
| Lives alone | 178 | 71 | 55.1 | 0.97 | (0.67 to 1.41) | 1.00 | (0.67 to 1.47) | 0.96 | (0.65 to 1.43) |
| Lives with non-relatives only | 101 | 101 | 57.1 | 1.05 | (0.64 to 1.72) | 1.12 | (0.67 to 1.86) | 1.29 | (0.76 to 2.17) |
| Lives with a partner, not with children | 171 | 114 | 38.2 | 0.49 | (0.34 to 0.70) | 0.46 | (0.31 to 0.70) | 0.46 | (0.31 to 0.69) |
| Lives with a partner and children | 149 | 90 | 50.3 | 0.80 | (0.57 to 1.14) | 0.77 | (0.52 to 1.14) | 0.73 | (0.49 to 1.10) |
| Lives with children, not with a partner | 246 | 94 | 66.2 | 1.55 | (1.12 to 2.14) | 1.64 | (1.16 to 2.31) | 1.54 | (1.08 to 2.21) |
| Attended a GUM clinic in the past year | | | | p<0.001 | | p<0.001 | | p<0.001 | |
| Lives with parents or other relatives | 860 | 479 | 26.8 | 1.00 | | 1.00 | | 1.00 | |
| Lives alone | 174 | 69 | 23.8 | 0.85 | (0.57 to 1.28) | 0.89 | (0.59 to 1.36) | 0.84 | (0.55 to 1.28) |
| Lives with non-relatives only | 100 | 100 | 26.5 | 0.98 | (0.52 to 1.85) | 1.05 | (0.55 to 2.00) | 1.25 | (0.65 to 2.40) |
| Lives with a partner, not with children | 163 | 109 | 12.2 | 0.38 | (0.23 to 0.61) | 0.39 | (0.23 to 0.68) | 0.40 | (0.23 to 0.69) |
| Lives with a partner and children | 149 | 90 | 8.2 | 0.25 | (0.13 to 0.45) | 0.26 | (0.14 to 0.49) | 0.24 | (0.13 to 0.45) |
| Lives with children, not with a partner | 239 | 91 | 21.5 | 0.75 | (0.48 to 1.16) | 0.80 | (0.51 to 1.26) | 0.74 | (0.46 to 1.18) |

Denominators vary across outcomes due to item non-response.

*Adjusted for age and relationship status.

†Adjusted for age, relationship status, employment/student status and area of residence.

‡ Opposite and/or same-sex partners.

§Reported two or more partners of the opposite or same sex with no condom use in the past year, or no condom use at first sex with one or more new partners of the opposite or same sex in the past year.

¶Diagnosed with chlamydia, gonorrhoea, herpes, genital warts, trichomoniasis, non-specific or non-gonococcal urethritis, or syphilis.

GUM, genitourinary medicine; STI, sexually transmitted infection.

**Table 4** Sexual risk behaviours and sexual health outcomes in sexually experienced men aged 16–24 years

| | Denominator (unweighted) | Denominator (weighted) | Prevalence (%) | OR | 95% CI | Partially adjusted OR* | 95% CI | Fully adjusted OR† | 95% CI |
|---|---|---|---|---|---|---|---|---|---|
| Any sexual intercourse in past 4 weeks | | | | p<0.001 | | p=0.24 | | p=0.49 | |
| Lives with parents or other relatives | 922 | 634 | 67.3 | 1.00 | | 1.00 | | 1.00 | |
| Lives alone | 154 | 81 | 77.3 | 1.66 | (1.10 to 2.50) | 1.34 | (0.78 to 2.31) | 1.41 | (0.82 to 2.43) |
| Lives with non-relatives only | 103 | 127 | 64.6 | 0.89 | (0.56 to 1.39) | 0.74 | (0.44 to 1.24) | 0.97 | (0.56 to 1.67) |
| Lives with a partner, not with children | 89 | 79 | 94.9 | 9.00 | (2.38 to 34.08) | 1.65 | (0.42 to 6.47) | 1.64 | (0.42 to 6.45) |
| Lives with a partner and children | 69 | 55 | 96.8 | 14.58 | (3.52 to 60.42) | 2.68 | (0.62 to 11.59) | 2.47 | (0.57 to 10.70) |
| Four or more occasions of sexual intercourse in the past 4 weeks | | | | p<0.001 | | p=0.77 | | p=0.50 | |
| Lives with parents or other relatives | 909 | 627 | 36.9 | 1.00 | | 1.00 | | 1.00 | |
| Lives alone | 149 | 78 | 49.2 | 1.66 | (1.04 to 2.65) | 1.15 | (0.55 to 2.40) | 1.19 | (0.60 to 2.35) |
| Lives with non-relatives only | 101 | 125 | 43.3 | 1.31 | (0.85 to 2.01) | 1.13 | (0.70 to 1.84) | 1.43 | (0.86 to 2.36) |
| Lives with a partner, not with children | 86 | 77 | 76.5 | 5.58 | (3.18 to 9.80) | 1.48 | (0.81 to 2.72) | 1.45 | (0.79 to 2.66) |
| Lives with a partner and children | 67 | 53 | 68.8 | 3.77 | (2.17 to 6.57) | 1.00 | (0.55 to 1.82) | 0.94 | (0.52 to 1.70) |
| Two or more partners‡ in the past year | | | | p<0.001 | | p<0.001 | | p<0.001 | |
| Lives with parents or other relatives | 932 | 641 | 48.9 | 1.00 | | 1.00 | | 1.00 | |
| Lives alone | 149 | 79 | 50.8 | 1.08 | (0.75 to 1.56) | 1.20 | (0.79 to 1.83) | 1.31 | (0.84 to 2.05) |
| Lives with non-relatives only | 105 | 129 | 43.1 | 0.79 | (0.51 to 1.24) | 0.81 | (0.52 to 1.25) | 1.06 | (0.67 to 1.68) |
| Lives with a partner, not with children | 89 | 79 | 11.9 | 0.14 | (0.07 to 0.29) | 0.21 | (0.10 to 0.47) | 0.21 | (0.10 to 0.46) |
| Lives with a partner and children | 69 | 55 | 2.8 | 0.03 | (0.01 to 0.13) | 0.05 | (0.01 to 0.19) | 0.04 | (0.01 to 0.18) |
| Had unsafe sex in the past year§ | | | | p=0.006 | | p=0.02 | | p=0.009 | |
| Lives with parents or other relatives | 931 | 642 | 31.3 | 1.00 | | 1.00 | | 1.00 | |
| Lives alone | 150 | 79 | 33.9 | 1.12 | (0.71 to 1.77) | 1.14 | (0.72 to 1.81) | 1.27 | (0.81 to 1.98) |
| Lives with non-relatives only | 105 | 129 | 26.6 | 0.79 | (0.49 to 1.29) | 0.80 | (0.49 to 1.30) | 1.17 | (0.69 to 1.99) |
| Lives with a partner, not with children | 89 | 79 | 25.3 | 0.74 | (0.41 to 1.33) | 0.81 | (0.44 to 1.49) | 0.84 | (0.45 to 1.55) |
| Lives with a partner and children | 69 | 55 | 9.6 | 0.23 | (0.10 to 0.52) | 0.25 | (0.11 to 0.59) | 0.25 | (0.11 to 0.57) |
| Diagnosed with an STI in the past year¶ | | | | p=0.92 | | p=0.99 | | p=0.98 | |
| Lives with parents or other relatives | 935 | 643 | 3.1 | 1.00 | | 1.00 | | 1.00 | |
| Lives alone | 155 | 82 | 2.9 | 0.94 | (0.37 to 2.37) | 0.96 | (0.37 to 2.51) | 1.00 | (0.37 to 2.67) |
| Lives with non-relatives only | 103 | 126 | 3.0 | 0.95 | (0.27 to 3.40) | 0.94 | (0.25 to 3.44) | 1.22 | (0.28 to 5.31) |
| Lives with a partner, not with children | 89 | 79 | 1.8 | 0.58 | (0.13 to 2.59) | 0.77 | (0.15 to 3.87) | 0.75 | (0.15 to 3.69) |
| Lives with a partner and children | 69 | 55 | 0.0 | n/a | n/a | n/a | n/a | n/a | n/a |
| Tested for chlamydia in the past year | | | | p=0.03 | | p=0.13 | | p=0.10 | |
| Lives with parents or other relatives | 941 | 648 | 37.1 | 1.00 | | 1.00 | | 1.00 | |

Continued

**Table 4** Continued

| | Denominator (unweighted) | Denominator (weighted) | Prevalence (%) | OR | 95% CI | Partially adjusted OR* | 95% CI | Fully adjusted OR† | 95% CI |
|---|---|---|---|---|---|---|---|---|---|
| Lives alone | 156 | 82 | 35.2 | 0.92 | (0.61 to 1.39) | 0.99 | (0.65 to 1.52) | 0.99 | (0.64 to 1.54) |
| Lives with non-relatives only | 106 | 130 | 34.2 | 0.88 | (0.57 to 1.35) | 0.95 | (0.62 to 1.48) | 1.05 | (0.65 to 1.68) |
| Lives with a partner, not with children | 89 | 79 | 25.1 | 0.57 | (0.35 to 0.93) | 0.62 | (0.37 to 1.07) | 0.62 | (0.36 to 1.05) |
| Lives with a partner and children | 69 | 55 | 20.3 | 0.43 | (0.23 to 0.81) | 0.47 | (0.24 to 0.92) | 0.47 | (0.24 to 0.91) |
| Attended a GUM clinic in the past year | | | | p=0.53 | | p=0.64 | | p=0.45 | |
| Lives with parents or other relatives | 925 | 637 | 16.8 | 1.00 | | 1.00 | | 1.00 | |
| Lives alone | 155 | 82 | 16.7 | 0.99 | (0.57 to 1.73) | 1.10 | (0.62 to 1.95) | 1.16 | (0.66 to 2.05) |
| Lives with non-relatives only | 106 | 130 | 19.2 | 1.18 | (0.69 to 2.03) | 1.28 | (0.73 to 2.24) | 1.50 | (0.82 to 2.75) |
| Lives with a partner, not with children | 89 | 79 | 16.7 | 0.99 | (0.54 to 1.81) | 1.29 | (0.67 to 2.49) | 1.29 | (0.67 to 2.49) |
| Lives with a partner and children | 66 | 52 | 9.4 | 0.51 | (0.23 to 1.15) | 0.67 | (0.28 to 1.57) | 0.64 | (0.27 to 1.50) |

Denominators vary across outcomes due to item non-response. Due to low numbers (n=4), statistics on sexual health outcomes and sexual risk behaviours are not presented for young men living with children but not with a partner.

*Adjusted for age and relationship status.

†Adjusted for age, relationship status, employment/student status and area of residence.

‡Opposite and/or same-sex partners.

§Reported two or more partners of opposite or same sex with no condom use in the past year, or no condom use at first sex with one or more new partners of the opposite or same sex in the past year.

¶Diagnosed with chlamydia, gonorrhoea, herpes, genital warts, trichomoniasis, non-specific or non-gonococcal urethritis, or syphilis.

GUM, genitourinary medicine; STI, sexually transmitted infection.

household structure. Using household structure as an indicator of a young person's life stage thus provides insight into the heterogeneity in development and lifestyle that exists for this age group, which should be taken into account when investigating young people's behaviour and health.

## Implications for future work

Overall, our study suggests that household structure, as a contextual demographic characteristic, might have an important role in understanding observed differences in sexual risk behaviours in young women, while playing less of a role for young men. Researchers might therefore consider incorporating household structure information into investigations of the sexual health and behaviour of young people. While this paper focused on young people, it is plausible that household structure may also play a role in sexual behaviour at other life stages. Given the heterogeneity in behaviour that exists across the six household structures we identified, our research also highlights the importance of sampling from a wide range of settings when studying the sexual behaviour and sexual health of young people.

In finding that women who live alone or with non-relatives report higher levels of sexual risk behaviours but not greater usage of sexual health services, our study suggests that young women living outside of a family unit, neither with parents nor a partner, might represent a group with a mismatch between STI risk and sexual healthcare/screening uptake. Clinicians should therefore perhaps consider household structure along with other contextual factors when assessing a young person's risk of STIs.

## Strengths and limitations

A major strength of this study is that, to our knowledge, it is the first to directly investigate the association between household structure and sexual risk behaviour and sexual health in young people. In doing so, we used individual-level data from a nationally representative sample of the British population, in a survey that captured a broad range of sexual health and behaviour outcomes.

While Natsal-3 interviewed over 15 000 women and men aged 16–74 years across Britain and oversampled young people resulting in 2140 women and 1729 men aged 16–24 years, it did not oversample ethnic minorities as in Natsal-2.[32] As a reflection of Britain's ethnic composition, the number of participants of non-white British ethnicity was therefore small, and so it was not feasible to adjust for ethnicity, which has been shown to be associated with household structure[33] and sexual behaviour,[34] and might therefore represent a confounder that we were unable to account for in our analyses. Natsal-3 did not collect information on participants' country of birth, an indicator of migrant status which is also associated with household structure.[3] Due to Natsal's sampling frame, the data are representative of young people living in private households, however young people living at university halls of residence, in institutions or who are homeless are likely to be under-represented. Although Natsal's study design, including the use of CASI, is specifically designed to minimise social desirability bias, it remains possible that the presence of parents or other household members at the time of interview may have influenced young participants' willingness to provide accurate answers to sensitive questions.[35 36] Though young people in Britain are most impacted by STIs,[16] the rates of diagnosed STIs at a population level, and therefore the absolute number of reported diagnoses in our sample, are relatively low. We therefore lacked statistical power to determine differences in reporting prior STI diagnoses by household structure with sufficient precision.

## CONCLUSIONS

Economic and social changes over the last 20 years have led to changes in the living situations of young people in Britain. Our study suggests that household structure may play a role in influencing their sexual risk behaviour. Future research should therefore take account of household structure to further our understanding of the heterogeneity of sexual behaviour, and by implication, sexual health of this key population group.

**Acknowledgements** The authors thank the study participants, the team of interviewers from NatCen Social Research, operations and computing staff from NatCen Social Research and the study funders.

**Contributors** This paper was conceived by TJC, NF, SC and CHM. TJC wrote the first draft of the article, with further contributions from NF, SC and CHM. TJC carried out the statistical analysis, with support from SC, NF and CHM. All authors interpreted data, reviewed successive drafts and approved the final version of the article.

**Funding** The study was supported by grants from the Medical Research Council (G0701757 and MR/N013867/1); and the Wellcome Trust (084840), with contributions from the Economic and Social Research Council and Department of Health. Soazig Clifton was funded to undertake independent research supported by the National Institute for Health Research (NIHR Research Methods Programme, Fellowships and Internships, NIHR-RMFI-2014-05-28).

**Disclaimer** The sponsors of the study had no role in study design, data collection, data analysis, data interpretation or writing of the report. The corresponding author had full access to all of the data in the study and had final responsibility for the decision to submit for publication.

**Competing interests** None declared.

**Patient consent** Not required.

**Ethics approval** The Natsal-3 study was approved by the Oxfordshire Research Ethics Committee A (reference: 09/H0604/27).

**Provenance and peer review** Not commissioned; externally peer reviewed.

**Data sharing statement** The Natsal-3 data set is publicly available from the UK Data Service: https://discover.ukdataservice.ac.uk/; SN: 7799; persistent identifier: 10.5255/UKDA-SN-77991-1.

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
