## [Reviewer comments · BMJ Open]

ARTICLE DETAILS

TITLE (PROVISIONAL)	Household structure and its association with sexual risk behaviours and sexual health outcomes: Evidence from a British probability sample survey
AUTHORS	Curtis, Tyrone; Field, Nigel; Clifton, Soazig; Mercer, Catherine

VERSION 1 – REVIEW

REVIEWER	Somefun Dolapo Oluwaseyi University of Witwatersrand, Johannesburg, South Africa
REVIEW RETURNED	06-Jun-2018

GENERAL COMMENTS	This paper is an important contribution in the public health field but requires more in-depth reflection of findings, that includes current narratives about adolescent sexual behaviour. I recommend the following revisions for this paper to be accepted: Abstract The hypothesis stated in the abstract is misleading. Are you investigating Household structure and sexual health care needs or sexual behaviour? Conclusion is a repetitive in what was presented in the result. Perhaps, give a policy implication regarding the findings. Introduction This section has been well written as it gives reasons as to the changing living arrangements among young adults in Britain. However, there are no references in the attached manuscript and I do not agree with the authors on a paucity of research in household structure and adolescent sexual behaviour in Britain and countries in close proximity. I believe that some studies may have defined household structure differently. I have suggested some studies for further reading just in case authors didn't use them. 1. Factors associated with self-reported first sexual intercourse in Scottish adolescents2. Socioeconomic Disadvantage and Adolescent Women's Sexual and Reproductive Behaviour: The Case of Five Developed Countries3. Influence of family type and parenting behaviours on teenage sexual behaviour and conceptions I also suggest that the paper is introduced against the backdrop of current and changing family structures in relation to adolescent sexual behaviour and not the information regarding the National survey. The introduction could be strengthened by briefly mentioning other determinants of adolescent sexual behaviour in Britain and a theoretical framework as to why household structure is significant in the lives of adolescents. This framework would also help understand the mechanisms through which household structure may influence
--

	adolescent sexual behaviour. Data and methods Was there any missing data in the study and how was this treated? Can you provide justification on the selected outcome variables? For instance, why have you defined recent sexual activity as “sex in the last four weeks”. This applies to all the outcome variables that have been used. Why have you combined living with parents and living with other relative together? Finally, what prompted the selection of the confounders used? Results When authors say “living with children”, which children are they referring to? Is it presence of children in the household or are they describing the living arrangement of other respondents in the household. If yes, why are they reporting the living arrangement of older persons in the household? The paper is based on young adults and I find it unnecessary. The use of “less likely” in presenting the descriptive results may be misleading. The discussion of bivariate associations would benefit from elaboration regarding association with control variables. If there are significant associations with selected confounders, this may limit the explanatory power of household structure and adolescent sexual behaviour. It is therefore relevant to discuss which confounders are more or less strongly correlated with adolescent sexual behaviour. This should apply in the adjusted results also. Discussion and Conclusion This section has been well written but it would be good to see further explanation as to why gender differentials exist.
--	---

REVIEWER	Jami Leichter Centers for Disease Control and Prevention, United States
REVIEW RETURNED	08-Jun-2018

GENERAL COMMENTS	This study uses national data from Britain to examine household structure (16-74 year olds) and its association with sexual health (16-24 year olds). It is interesting and is an addition to the literature base. I have a few suggestions for improvement.  • Please add the response rate for the Natsal-3. • Page 4, lines 30-37 & page 5, lines 13-20: The introduction sets the context for the study very well. However, the issue of changing social/economic context suggests that a higher age range may be useful for this study. Perhaps up to age 29? While < 24 is certainly at higher STI risk (though this isn't a main outcome for this study), 25-29 is also at risk and it would be interesting given that the changing social/economic context has likely changed their living arrangements more than it has for 16-24. Also, this context is likely relevant for many higher income countries post-recession; thus, it may be useful to increase the age range for this subset of analyses. Finally, the title focuses on sexual health rather than just STI so an expansion of the age range would fit under this framework. • Page 6, it would be helpful to label the variables described as “predictors” and “outcomes” or something similar. Also, some of the variables aren't mentioned until the statistical analyses or the tables.  o Do you have any measure of income (poverty/class)? If so, it would be good to include given the introduction. • Recommend adding some justification or support for at least outcomes by group. There are a lot of them and without this the analysis feels exploratory.
--

	 • Also, I do not recommend talking about sexual behavior and sexual health outcomes separately. Behavior, like health care use/receipt of services and disease, is all a part of sexual health. • Regarding behavior, it is best if you can stick with one term throughout the paper (methods, results, discussion) – sexual risk or unsafe sex. I would vote for sexual risk.
--	---

REVIEWER	Dr. Daniel Oshi The University of the West Indies, Mona Campus, Jamaica.
REVIEW RETURNED	24-Jul-2018

GENERAL COMMENTS	General Comments This is a well-written manuscript addressing a social issue that is very crucial, that is, sexual behaviour of young persons and its association with household structure. The study is also timely considering the current shifts in the household living patterns in Britain. Title and Abstract The title is appropriate, and included the study design. The abstract is informative and contains a balanced summary of the methods, results and conclusions. Introduction This presents a clear and adequate background to the research problem, with good coverage of existing knowledge of the issue in the British context, using current literature (most being less than ten years old). The authors clearly established a clear gap in knowledge, and stated how their research would contribute to filling this gap (good justification). Methods The authors made reference to two sources where more details of the methods had already been published. The research design and the sampling techniques were succinctly described. Study setting, source population and the period of study were stated. Adequate description of questionnaire administration was given. Key concepts were clearly defined while the main variables were specified. Clear description was given of how the variables were handled in the analysis, including the demographic group (16-24 years) that their analysis focused on, and why. The authors adequately described how they carried out univariate and multivariable analysis, the choices they made and why. However, 'relationship status' does not appear to be clearly defined/ explained. It is not clear whether the authors used household 'partners' as proxy indicators for 'relations/ relationships'; but this is not likely. Since relationship status was one of the potential confounding characteristics, it will be helpful for the authors to unambiguously define it. Results Very well written with due attention to reporting of numbers at each
--

	stage of the analysis. However, one does not get a clear picture of the socio-demographic characteristics of the participants (except age, gender and household structure). Other key characteristics such as relationship status, area of residence and employment/ student status were missing at the level of descriptive analysis, despite the fact that the authors later 'fed' these characteristics into their multivariable analysis by treating them (correctly!) as potential confounders. Discussion Well written, and implications for future research as well as strengths and limitations included. Conclusion The conclusion seems to relate more to the role of household information in survey data analysis than to the actual findings of the study. I suggest that the authors re-phrase their conclusion to make it more directly focused on the findings of their study rather than on the methodological issues surrounding the use of household structure information. It seems to me that the study was not about the 'usefulness' of household structure information but about the association between household structure and sexual health and sexual behaviour, as the authors themselves stated in the last paragraph of their Introduction. Therefore, their conclusion should be so drawn to focus sharply on this association and its societal implication. Recommendation Accept with minor revision
--	---

VERSION 1 – AUTHOR RESPONSE

Reviewer: 1

Abstract

1. The hypothesis stated in the abstract is misleading. Are you investigating Household structure and sexual health care needs or sexual behaviour?

Response: We stated our hypothesis as written as we are investigating sexual health in its broadest sense, thus we have included both sexual risk behaviours and sexual health care outcomes in our hypothesis – as highlighted by Reviewer 2 (comment 6).

2. Conclusion is a repetitive in what was presented in the result. Perhaps, give a policy implication regarding the findings.

Response: We thank the reviewer for their suggestion, and have now rewritten the abstract conclusion to focus more on the implications of our research.

Introduction

3. This section has been well written as it gives reasons as to the changing living arrangements among young adults in Britain. However, there are no references in the attached manuscript and I do not agree with the authors on a paucity of research in household structure and adolescent sexual behaviour in Britain and countries in close proximity. I believe that some studies may have defined household structure differently. I have suggested some studies for further reading just in case authors didn't use them.

1. Factors associated with self-reported first sexual intercourse in Scottish adolescents
2. Socioeconomic Disadvantage and Adolescent Women's Sexual and Reproductive Behaviour: The Case of Five Developed Countries
3. Influence of family type and parenting behaviours on teenage sexual behaviour and conceptions

Response: We were pleased to read that the Reviewer found the Introduction well-written. Most previous research on household structure and young adults' sexual behaviour has focused specifically on *family* structure, i.e. the presence of one or both parents in the house, and has typically focused on young people below the age of 18. This has now been mentioned in our Introduction. In contrast, and thus our paper's unique contribution, is that we investigate household structures more broadly than just in the context of family structure, reflecting our paper's focus on the sexual health of young *adults* aged 16-24, in whom the burden of sexual ill-health is greatest.

4. I also suggest that the paper is introduced against the backdrop of current and changing family structures in relation to adolescent sexual behaviour and not the information regarding the National survey.

Response: Thank you for this suggestion, we have now restructured the Introduction.

5. The introduction could be strengthened by briefly mentioning other determinants of adolescent sexual behaviour in Britain and a theoretical framework as to why household structure is significant in the lives of adolescents. This framework would also help understand the mechanisms through which household structure may influence adolescent sexual behaviour.

Response: We have now added references to previous studies looking at family structure as a predictor of adolescent sexual behaviour, as well as a study that applied the integrative model of behavioural prediction framework to adolescent sexual behaviour [p.4]. We also acknowledge that one of our paper's unique contributions is its focus on older young people vs. adolescents.

Data and methods

6. Was there any missing data in the study and how was this treated?

Response: The use of CAPI and CASI to deliver the survey resulted in very little missing data, with non-response rates typically below 0.5% for CAPI questions and between 1-3% for CASI questions, as previously reported (reference #18). Respondents with missing data for individual outcomes were dropped from the analyses, but there were relatively few of these in our study. The outcome with the most missing data was "four or more occasions of sex in past 4 weeks", for which no data are available for 147 (8.6% of sexually-experienced young women) and 53 (3.9% of sexually-experienced young men); rates of missing data for all other outcomes were much lower.

7. Can you provide justification on the selected outcome variables? For instance, why have you defined recent sexual activity as "sex in the last four weeks". This applies to all the outcome variables that have been used.

Response: As we have now explained in our Methods [p.7], sexual health is a multidimensional concept that cannot be adequately captured using a single indicator. A strength of our paper is its inclusion of a number of different measures of sexual health, which provide nuanced insights into this dimension of health. The timeframes used for the behaviours and outcome variables reflect the need to achieve a balance between too recent a timeframe, which leads to too few participants reporting a given outcome (especially for relatively uncommon outcomes such as testing for chlamydia), and too long a time frame, which loses recency and therefore precision. The latter point is particularly relevant for young people for whom there is greater fluctuation over time relative to older people.

8. Why have you combined living with parents and living with other relative together?

Response: Categorising household structure is complex and necessarily involves collapsing some less prevalent groups in order to achieve a workable number of categories containing a workable number of participants who can be reasonably grouped together. The decision here was in recognition of the fact that some young people may not live with their parents, for a variety of reasons, and may instead be living with other caretakers such as grandparents or even older siblings, and their relationship to these caretakers may have a similar influence on behaviour. Furthermore, the number of respondents living with other relatives but not with parents was not large enough to treat them as a

separate category, and so they were grouped with those living with their parents. We have now included the reasoning behind our decision as a footnote to Table 1 [p. 6].

9. Finally, what prompted the selection of the confounders used?

As we now discuss in the Methods [p.8], these are variables thought to be associated with both household structure and sexual health. Area of residence is likely to affect availability of partners as well as living situation due to costs. Age is obviously associated with both, as is relationship status. Employment/student status is also likely to be associated with household structure, but also possibly with sexual behaviour. It was not possible to include ethnicity in our analyses due to the small numbers in some categories which made the model unstable, as discussed in the limitations section.

Results

10. When authors say “living with children”, which children are they referring to? Is it presence of children in the household or are they describing the living arrangement of other respondents in the household. If yes, why are they reporting the living arrangement of older persons in the household? The paper is based on young adults and I find it unnecessary.

As stated in our Methods, “children” or “grandchildren” refer to children and grandchildren of the participant, including step-, foster or -in-law. Young siblings would be considered as other relatives, while children who are not related to the respondent would be considered to be non-relatives. We have added text to Table 1 [p.6] to further clarify this issue.

11. The use of “less likely” in presenting the descriptive results may be misleading.

Response: Thank you, this has now been changed throughout the descriptive results.

12. The discussion of bivariate associations would benefit from elaboration regarding association with control variables. If there are significant associations with selected confounders, this may limit the explanatory power of household structure and adolescent sexual behaviour. It is therefore relevant to discuss which confounders are more or less strongly correlated with adolescent sexual behaviour. This should apply in the adjusted results also.

Response: Much research has previously been carried out examining associations between the variables we consider as control variables in this paper, such as relationship status and sexual behaviour, and we now cite these studies. In order to demonstrate more clearly to the reader how the key confounders identified affect the hypothesised association between household structure and indicators of sexual health, which is the focus of our paper, we now present in Tables 3 and 4 partially adjusted ORs (adjusting for age and relationship status, which are the confounders most strongly correlated with sexual behaviour), as well as the fully adjusted ORs (adjusting for all confounders). We now explain this in the Statistical analysis section of the paper [p.8].

Discussion and Conclusion

13. This section has been well written but it would be good to see further explanation as to why gender differentials exist.

Response: We have now added a brief discussion about differences between men and women in the Discussion.

Reviewer: 2

This study uses national data from Britain to examine household structure (16-74 year olds) and its association with sexual health (16-24 year olds). It is interesting and is an addition to the literature base. I have a few suggestions for improvement.

1. Please add the response rate for the Natsal-3.

Response: This has now been added in the Methods section.

2. Page 4, lines 30-37 & page 5, lines 13-20: The introduction sets the context for the study very well. However, the issue of changing social/economic context suggests that a higher age range may be useful for this study. Perhaps up to age 29? While < 24 is certainly at higher STI risk (though this isn't a main outcome for this study), 25-29 is also at risk and it would be interesting given that the

changing social/economic context has likely changed their living arrangements more than it has for 16-24. Also, this context is likely relevant for many higher income countries post-recession; thus, it may be useful to increase the age range for this subset of analyses. Finally, the title focuses on sexual health rather than just STI so an expansion of the age range would fit under this framework. Response: We agree with the Reviewer that the issues addressed in our paper are increasingly affecting older younger adults, not just those in their late 20s but sometimes well into their 30s. However, we feel that the age range of 16-24y is most relevant given that many sexual and reproductive health policies still focus on those aged under 25y.

3. Page 6, it would be helpful to label the variables described as “predictors” and “outcomes” or something similar. Also, some of the variables aren’t mentioned until the statistical analyses or the tables.

Response: Thank you for this suggestion, this has now been incorporated into the paper. In addition, we have listed all outcomes studied in the Methods section [p. 7].

4. Do you have any measure of income (poverty/class)? If so, it would be good to include given the introduction.

Response: For the first time, Natsal-3 asked participants about their household income, but this question had (relatively) high item non-response (~20%). We were concerned that this would introduce potential bias to our analysis. We were also concerned about measurement error, given that young people living with their parents or someone sharing with non-family members might not know or be able to precisely estimate their household income. On this basis, this variable was not used in the present analysis. However, we do include data on participant’s social class according to the National Statistics Socio-Economic Classification, which we used to categorise participants as student, in employment, or not in employment/training in the later analyses in the paper. This attempts to take account of variation by class, whilst acknowledging that assessment of class / wealth status is particularly difficult for young adults who are often still in the process of establishing their careers (see Sheringham J et al. 2013). This detail is now added to the Methods [p.8].

5. Recommend adding some justification or support for at least outcomes by group. There are a lot of them and without this the analysis feels exploratory.

Response: We thank the reviewer for this suggestion, which mirrored a similar request from Reviewer 1. We have now added to the Methods a brief explanation of why we have chosen the outcomes presented here [p.7].

6. Also, I do not recommend talking about sexual behavior and sexual health outcomes separately. Response: Sexual behaviour, health care use, and sexual health outcomes are all components of sexual health. We felt it was important to separate these in this paper in part reflecting how the latter often result from the former. This is particularly important when thinking about how interventions may operate.

7. Regarding behavior, it is best if you can stick with one term throughout the paper (methods, results, discussion) – sexual risk or unsafe sex. I would vote for sexual risk.

Response: Thank you for this comment. In this paper the term “unsafe sex” refers to a specific outcome defined as reporting two or more partners of opposite or same sex with no condom use in the past year, or no condom use at first sex with one or more new partners of opposite or same sex in the past year. In contrast, sexual risk behaviour is used as a broader “umbrella term” to capture behaviours that puts individuals at risk of poor sexual health outcomes. We have therefore opted to retain the use of “unsafe sex” as an outcome name but have used the term “sexual risk behaviour” with more consistency throughout the paper.

Reviewer: 3

General Comments

1. This is a well-written manuscript addressing a social issue that is very crucial, that is, sexual behaviour of young persons and its association with household structure. The study is also timely considering the current shifts in the household living patterns in Britain.

Response: We thank Reviewer 3 for recognising the importance of our paper.

Title and Abstract

2. The title is appropriate, and included the study design. The abstract is informative and contains a balanced summary of the methods, results and conclusions.

Response: We again thank Reviewer 3 for their positive feedback.

Introduction

3. This presents a clear and adequate background to the research problem, with good coverage of existing knowledge of the issue in the British context, using current literature (most being less than ten years old). The authors clearly established a clear gap in knowledge, and stated how their research would contribute to filling this gap (good justification).

Response: We thank the reviewer for their positive comments.

Methods

4. The authors made reference to two sources where more details of the methods had already been published. The research design and the sampling techniques were succinctly described. Study setting, source population and the period of study were stated. Adequate description of questionnaire administration was given. Key concepts were clearly defined while the main variables were specified. Clear description was given of how the variables were handled in the analysis, including the demographic group (16-24 years) that their analysis focused on, and why. The authors adequately described how they carried out univariate and multivariable analysis, the choices they made and why. However, 'relationship status' does not appear to be clearly defined/ explained. It is not clear whether the authors used household 'partners' as proxy indicators for 'relations/ relationships'; but this is not likely. Since relationship status was one of the potential confounding characteristics, it will be helpful for the authors to unambiguously define it.

Response: Thank you for this comment. We have now added a brief description of how the relationship status variable was derived to the Methods section [p.8]. With regards to the link between the relationship status variable and household structure, these are separate but related variables in that a respondent could only be categorised as living with a partner (with or without children) if they reported also currently being in a relationship (married, in a civil partnership or in a committed relationship). The questions asked to participants about members of their household and about their marital status can be found under the Household Classification section [p. 105] of the Natsal-3 questionnaire (<http://www.natsal.ac.uk/media/2078/b1-capi-and-casi-questionnaire.pdf>).

Results

5. Very well written with due attention to reporting of numbers at each stage of the analysis. However, one does not get a clear picture of the socio-demographic characteristics of the participants (except age, gender and household structure). Other key characteristics such as relationship status, area of residence and employment/ student status were missing at the level of descriptive analysis, despite the fact that the authors later 'fed' these characteristics into their multivariable analysis by treating them (correctly!) as potential confounders.

Response: Thank you, we recognise the epidemiological importance of describing the key demographic characteristics of the sample, and we have added a table to the manuscript [Table 2], showing these characteristics for the sample as a whole and also showing for each of the different types of household structure.

Discussion

6. Well written, and implications for future research as well as strengths and limitations included.

Response: Thank you.

Conclusion

7. The conclusion seems to relate more to the role of household information in survey data analysis than to the actual findings of the study. I suggest that the authors re-phrase their conclusion to make it more directly focused on the findings of their study rather than on the methodological issues surrounding the use of household structure information. It seems to me that the study was not about the 'usefulness' of household structure information but about the association between household structure and sexual health and sexual behaviour, as the authors themselves stated in the last paragraph of their Introduction. Therefore, their conclusion should be so drawn to focus sharply on this association and its societal implication.

Response: We thank the reviewer for their comment. We have now rewritten the conclusion to more accurately represent the focus of the paper.

VERSION 2 – REVIEW

REVIEWER	Jami Leichter Deputy Associate Director for Science, Division of STD Prevention, US
REVIEW RETURNED	05-Sep-2018

GENERAL COMMENTS	On page 9, please replace "multivariate" with "multivariable"
---

REVIEWER	Dr. Daniel C. Oshi The University of the West Indies, Mona, Jamaica
REVIEW RETURNED	28-Aug-2018

GENERAL COMMENTS	The authors have completely addressed the issues I raised in my review. I agree with the revisions and explanations/ responses provided by the authors.
---